# DNA Repair Inhibitors: Potential Targets and Partners for Targeted Radionuclide Therapy

**DOI:** 10.3390/pharmaceutics15071926

**Published:** 2023-07-11

**Authors:** Honoka Obata, Mikako Ogawa, Michael R. Zalutsky

**Affiliations:** 1Department of Advanced Nuclear Medicine Sciences, National Institutes for Quantum Science and Technology (QST), 4-9-1 Anagawa, Inage-ku, Chiba 263-8555, Japan; obata.honoka@qst.go.jp; 2Department of Molecular Imaging and Theranostics, National Institutes for Quantum Science and Technology (QST), 4-9-1 Anagawa, Inage-ku, Chiba 263-8555, Japan; 3Departments of Radiology and Radiation Oncology, Duke University Medical Center, Durham, NC 27710, USA; 4Graduate School of Pharmaceutical Sciences, Hokkaido University, Kita-ku, Sapporo 060-0812, Japan; mogawa@pharm.hokudai.ac.jp

**Keywords:** targeted radionuclide therapy, auger electron emitters, DNA damage response, DNA repair inhibitor, transcription factor, PARP

## Abstract

The present review aims to explore the potential targets/partners for future targeted radionuclide therapy (TRT) strategies, wherein cancer cells often are not killed effectively, despite receiving a high average tumor radiation dose. Here, we shall discuss the key factors in the cancer genome, especially those related to DNA damage response/repair and maintenance systems for escaping cell death in cancer cells. To overcome the current limitations of TRT effectiveness due to radiation/drug-tolerant cells and tumor heterogeneity, and to make TRT more effective, we propose that a promising strategy would be to target the DNA maintenance factors that are crucial for cancer survival. Considering their cancer-specific DNA damage response/repair ability and dysregulated transcription/epigenetic system, key factors such as PARP, ATM/ATR, amplified/overexpressed transcription factors, and DNA methyltransferases have the potential to be molecular targets for Auger electron therapy; moreover, their inhibition by non-radioactive molecules could be a partnering component for enhancing the therapeutic response of TRT.

## 1. Overview of the Current Status of Targeted Radionuclide Therapy (TRT)

Targeted radionuclide therapy (TRT) is a type of internal radiotherapy using a radionuclide-labeled agent that selectively delivers a radiation dose to targeted sites such as cancer cells. In TRT, targeted malignant tissues are irradiated and can be damaged efficiently with the short-range radiation (*β*^−^-particles, *α*-particles, and Auger electrons) emitted from the radionuclide. An ideal delivery system for the radionuclide enables targeting and treating even nondetectable cancer cells throughout the body. This unique treatment modality thus offers the exciting possibility of controlling metastatic disease, even at occult sites, with low invasiveness.

### 1.1. β^−^-Particle or α-Particle-Based TRT Exhibits Therapeutic Efficacy: On the Way to General Utility

*β*^−^-particles and *α*-particles have been applied in TRT [1]. *β*^−^-particles are the most commonly used in the clinic [1], exhibiting cytotoxicity for relatively large deposits of cancer cells because of their emission of energetic electrons of a moderate tissue range (0.05–12 mm) [2] (Figure 1). Typical examples approved by the US Food and Drug Administration (FDA) are ^89^Sr chloride ([^89^Sr]SrCl_2_) [3], ^90^Y ibritumomab tiuxetan (Zevalin) [4], and *meta*-[^131^I]iodobenzylguanidine ([^131^I]MIBG) [5]. In recent years, ^177^Lu has become popular because it emits relatively low-energy *β*^−^-particles (E_mean_ = 133.6 keV [6]), compared to other candidate radionuclides (E_mean_ = 181.9 keV for ^131^I, 763 keV for ^188^Re, 932.3 keV for ^90^Y, [6]). The FDA approved [^177^Lu]Lu-DOTATATE (Lutathera^®^) in 2018 [7] and [^177^Lu]Lu-PSMA-617 in 2022 [8]. The low-energy *β*^−^-emission being more likely to irradiate the targeted tumors than the surrounding normal tissues and the absence of moderate-energy gamma-ray emission, as found with ^131^I, are characteristics that continue to make ^177^Lu the most actively investigated *β*^−^-emitter for TRT.

Compared to *β*^−^-particles, *α*-particles have a higher linear energy transfer (LET) (80 keV/µm) and a shorter range in tissues (40–100 μm) [2], which can minimize the radiation dose to normal cells and maximize the doses to cancer cells, if they are properly targeted (Figure 1). Especially since the research group of Heidelberg University reported superior clinical results in some patients for [^225^Ac]Ac-PSMA-617 in metastatic castration-resistant prostate cancer [9], *α*-emitters have been attracting increased interest due to their higher therapeutic efficacy. Ra-223, Ac-225, At-211, Pb-212, and Bi-213 are promising *α*-emitters, and numerous agents labeled with these radionuclides are being developed [10]. Ra-223 dichloride ([^223^Ra]RaCl_2_; Xofigo^®^ (Bayer, Leverkusen, Germany)) was approved by the FDA in 2013 for bone-metastatic castration-resistant prostate cancer [11]. Ac-225 is popular, led by [^225^Ac]Ac-PSMA-617 [9], because of its high cell-killing potential due to its multiple *β*^−^/*α*-emission decay chain, although its production, availability, and in vivo stability are significant issues. At-211 is also a promising candidate because it can be produced using a cyclotron. This leads to an advantage in its potential availability and versatility, and some human studies are ongoing with ^211^At-labeled antibodies, [^211^At]NaAt, as well as others [12].

In TRT, its therapeutic effect increases when the tumor dose is elevated. Numerous therapy experiments have demonstrated a high therapeutic efficacy with a variety of agents and tumor types in various mouse xenograft models. However, the response rate has not been so high in patients; sometimes, even an FDA-approved drug has had no significant survival advantage compared to standard of care [13]. A typical explanation, inevitably, is that the calculated radiation tumor dose was not sufficient to kill all the cancer cells. Variable drug accumulation in different-sized tumors with a heterogeneous vascular distribution, along with tumor heterogeneity and treatment-resistant cancer cells, can reduce the likelihood of successful therapy. Unfortunately, it would be difficult to irradiate and kill all of the cancer cells without any adverse effects on normal tissues (Figure 1).

In order to improve its clinical efficacy and response rate, further drug development efforts and combination therapy strategies are needed if more effective TRT without compromising normal tissues is to be achieved. Furthermore, inflammation/immune responses might affect the results of clinical studies. Radiation is indeed known to activate the antitumor immune system via several pathways [14], and recent studies have demonstrated that a combination of radiotherapy and immune checkpoint inhibitors can provide a high therapeutic efficacy [15]. There might be some immune-related contribution in TRT patients with complete responses, although this remains unclear and, at present, uncontrollable. Combining TRT and immune checkpoint inhibitors is being further investigated and might improve TRT therapeutic results.

Thus, *β^−^*/*α*-particle TRT has been partially established at the clinical stage, as proven by many clinical studies and a few FDA-approved agents. However, TRT is not ready for widespread use, because it is necessary to both improve its effectiveness and expand its range of applicable cancer types. To accomplish this, it is essential not only to increase tumor radiation dose, but also to enhance the radiation sensitivity of the cancer cells selectively.

### 1.2. Auger-Electron TRT Remains Attractive but Success Is Elusive

Auger electrons, the third type of radiation relevant to TRT, are of a high LET (4–26 keV/µm) with a nano-scale range of action (2–500 nm) [2] (Figure 1). Auger electrons are capable of predominately damaging the targeted molecules in cancer cells and causing the least damage to the nontargeted normal cell populations. Hypothetically, this makes Auger electron emitters an ideal source for internal radiotherapy as, under a perfect drug delivery scenario, there would be minimal adverse normal tissue effects. There have been many attempts to develop TRT agents labeled with Auger-electron-emitting radionuclides, especially with ^125^I and ^111^In (Table 1 [16,17,18,19,20,21,22,23,24,25]) (e.g., [^125^I]IUdR [16], ^125^I-labeled huA33 [17], ^111^In-DTPA-octreotide [19], and ^111^In-DTPA-EGFR [21]). However, the therapeutic efficacy of these agents has been modest or low in the clinical trials performed to date. The causes of their limited usefulness and potential solutions to this behavior are under investigation. Meanwhile, the development of these and other Auger-electron TRT agents is ongoing.

One likely factor is that the range of Auger electrons is much smaller than a single cell. Since the range of *β*^−^-particles and *α*-particles covers at least a few cells, tumor cells can be killed by delivering more radionuclides and depositing higher radiation doses to the targeted tumor region, taking advantage of the physical crossfire effect. Therefore, an intracellular targeting strategy is not relevant to TRT approaches with *β*^−^-particle and *α*-particle emitters. However, the range of Auger electrons is extremely short, suggesting an additional barrier for therapy. To kill cells efficiently, it is necessary to transport the radiopharmaceutical to the intracellular regions that are sensitive to the radiation of Auger electrons [26]. Unfortunately, ideal drug delivery systems and therapeutic approaches for Auger-electron TRT have yet to be established.

DNA is generally considered to be the primary target for Auger electrons [27], because DNA damage exemplified by double-strand breaks (DSBs) caused by high-LET radiation is known as a major factor leading to cell death [28]. To cause cell death from DNA damage induced by Auger electrons to occur more efficiently, we consider three main factors (Figure 1). One is the distance between the DNA and the radionuclide emitting Auger electrons. According to some studies, Auger-electron-emitting radionuclides decaying closer to the DNA can induce more DSBs [29,30,31]. Thus, the delivery of the radionuclides as close as possible to the DNA is essential in order to maximize the interaction between the Auger electrons and DNA in a nano-scale range. The second consideration is the number of radionuclides delivered to the DNA. In the case of [^125^I]IUdR, the number of DNA-incorporated ^125^I decays required to cause cell death was estimated to be about 30–60 per cell [32]. These two factors are related to escalating the radiation dose given to the DNA from the TRT agent. Additionally, we propose a third consideration as a crucial strategy for Auger-electron TRT—targeting and damaging the specific sites of the genome that are involved in the survival mechanism in living cells. To summarize, to cause cell death more efficiently via critical DNA damage to cancer cells, more Auger-electron-emitting radionuclides need to directly bind to key DNA sites such as oncogenes, which are related to cancer cell survival mechanisms. If Auger electrons could induce damage predominately to these key targets within cancer cells, this would become an ideal “smart radiotherapy” without any adverse effects, which cannot be accomplished with *β^−^*/*α*-particles because of their multicellular tissue range.

Additionally, the cell cycle checkpoint and DNA repair systems could also be exploited after damaging DNA with Auger electrons. One can ask rhetorically: which would be a better strategy for Auger electron cancer therapy, (a) causing enough DNA damage beyond the DNA repair capacity to lead to cell death instantly after irradiation, or (b) inducing moderate DNA damage and keeping the cell cycle operant to lead to reproductive death after cell proliferation? Clearly, it would be inefficient to cause moderate DNA damage only for it to be repaired by activating the repair system. Therefore, the type of DNA damage and its induction rate should be reconsidered to be able to design the most efficient therapy approaches.

Hence, irrespective of the potential of Auger electrons for cancer treatment, Auger electron TRT has yet to achieve a practical and effective therapeutic response. It is still unclear what molecular target should be damaged by Auger electrons and what type of DNA damage is the most efficient at treating tumors. A further survey of the promising targets associated with DNA molecules and their functions is essential, and will be attempted in the subsequent sections of this review.

## 2. A New Therapeutic Strategy to Break through Current TRT Limitations: Making TRT More Effective

Current TRT approaches with radionuclides emitting *α*-particles and/or *β^−^*-particles have enjoyed some success, including FDA-approved products; however, at this point, their impact has yet to reach its potential at the clinical level. Although many studies have attempted to deliver more radioactive drugs to tumors to try to achieve a higher therapeutic effect, it is clearly impossible to deliver all of the injected drug to tumors. It is also difficult to distribute the drug to all the cells within a tumor, with the impact of heterogeneity being dependent on the tumor size and vascular distribution [33]. Cells lacking a sufficient dose to be killed remain, causing the tumor to regrow. These characteristics suggest the need for additional strategies to make TRT more effective beyond increasing the average tumor radiation dose.

Herein, we shall discuss three approaches that could be pursued for improving the success of TRT in the future: (1) expanding the difference in cytotoxicity towards normal cells and cancer cells; (2) increasing the therapeutic sensitivity of radiation-resistant cancer cells; and (3) attacking unirradiated cells indirectly using inflammatory/immune responses. In the next generation of TRT strategies designed to overcome its current limitations, it may be necessary to not only cause cell death with an escalated radiation dose, but also to direct damage to cancer-specific molecular systems, especially by focusing on the cancer genome. We hypothesize that targeting cancer-related genomic factors with the radioactive drug or combining appropriate inhibiting agents with TRT has the potential to overcome the current limitations of TRT.

Enhancing the therapeutic specificity between cancer cells and normal cells, exploiting their differences, especially in the DNA damage response and repair ability, would be important. In TRT, to make the cytotoxicity to cancer cells higher than that to normal cells, radiolabeled agents are targeted to the antigens, transporters, or receptors that are highly over-expressed on the cancer cell membrane [34]. This means that the delivery of more radionuclides to cancer cells than normal cells will create a higher therapeutic index in cancer therapy. However, we speculate that another way to make TRT more cancer-specific would be downregulating the cancer-specific genomic functions related to DNA damage response/repair. Cancer cells proliferate abnormally and at a much faster rate than normal cells while escaping cell death. While spontaneous DNA damage and replication errors are inevitable even in normal cell division [35], such abnormal DNA synthesis and cell division are also likely to put the cell at risk. Therefore, cancer cells develop a variety of degenerated genomic functions, including DNA damage response/repair. This contributes to an infinite proliferative capacity, which is characterized as genomic instability in cancer [36]. It is important for cancer cells to keep their genome structure and function, using many upregulated molecules while escaping critical DNA damage and/or programmed cell death by enhanced irregular signaling and repair [37,38]. Thus, such upregulated genomic factors and their functions could possibly be exploited as part of the therapeutic process, allowing future TRT to have an enhanced therapeutic specificity.

This knowledge can also be employed for increasing the therapeutic sensitivity of radiation-resistant cancer cells. Radiation resistance is derived from the genetic background and alterations of the key regulators. These radiation-tolerant cells cannot be killed efficiently by simply increasing the radiation dose. Alternatively, one could attempt to downregulate the specific molecular factors that are involved in the survival mechanisms that act to reduce damage. Copious studies have identified the potential targets for abnormal transcription and metabolism that are related to DNA damage response/repair (e.g., X-ray repair cross-complementing 1, XRCC1; replication protein A, RPA; Poly(ADP-ribose)polymerases, PARPs) and the cell cycle checkpoint (e.g., ataxia telangiectasia mutated, ATM) and signaling pathways [39]. Thus, therapeutic strategies targeting these crucial molecular factors that are overexpressed in radiation-tolerant cells may enhance their radiation sensitivity and help to overcome therapeutic resistance.

Additionally, it is challenging to control secondary signaling pathways and inflammatory/immune responses. Radiation therapy tends to be followed by cellular profile changes, including gene expression [40], signaling [41], and metabolism [42]. This variability seems complex and uncontrollable because conventional radiotherapy results in a high-energy deposition to whole cells. Radiation-induced DNA damage sometimes activates the immune/inflammatory response via gene expression alteration and subsequent changes in the signaling pathways. Immune activation after radiotherapy is known as the abscopal effect. This mechanism is expected to result from the cytotoxic T cells that are driven by the highly expressed chemokines/cytokines and tumor cell-derived antigens released during therapy [43]. Many research articles have investigated the abscopal mechanism using immune-competent mice bearing murine tumors (e.g., 4T1 breast cancer cells; Lewis lung carcinoma cells; and MC38 colorectal carcinoma cells) [44]. Some models have demonstrated the abscopal effect with radiotherapy using different dosing and fractionation strategies, and immunotherapy with immune checkpoint inhibitors such as anti-PD1/PDL1 and anti-CTLA4 [45,46]. Unfortunately, exploiting the abscopal effect in the clinic remains elusive, so it is currently not a reliable therapeutic regimen at the patient treatment level. On the other hand, if specific genomic factors and their functions could be damaged using TRT, it might be possible to control a subsequent damage response pathway that might activate the inflammatory/immune systems. Apart from that, the bystander effect in radiotherapy involves intercellular signaling between irradiated and non-irradiated cells via several secretion factors [47]. The bystander effect has also been involved in a variety of molecular signaling pathways [48]. If the bystander effect could be completely regulated, its damage-response signaling and immune/inflammatory activation could then be controlled. Hence, a secondary signaling pathway and inflammatory/immune response initiated by radiation would have the potential to exert an indirect therapeutic effect, in addition to the primary radiation-related effect of the TRT agent. This strategy might contribute to treating the whole tumor cell population, even with only a partial irradiation of the constituent cancer cells.

## 3. Potential Molecular Targets and Their Inhibitors as TRT Partners: Focusing on DNA Damage Response/Repair and Cell Cycle Maintenance Systems in Cancer Cells

In the cancer genome, there are several potential molecular targets controlling DNA damage response/repair, cell cycle checkpoints, and cell division. These molecules might be able to be attacked directly using proper targeting agents combined with Auger electron therapy. As noted earlier, Auger electrons can potentially cause critical damage to cells in the nano-scale range. Considering this property, Auger electrons could be used for damaging crucial molecular factors by precisely targeting them to downregulate their function. Selectively damaging the key regulatory molecules within cancer cells using Auger electrons should cause a higher cytotoxicity than randomly damaging DNA molecules.

With TRT agents that emit *α*-particles and/or *β^−^*-particles, it would be inefficient to try to damage these key regulatory molecules directly, because their energy is transferred over a millimeter to micrometer range. On the other hand, considering the effective range of *α*-particles and *β^−^*-particles, it might be reasonable to increase their therapeutic efficacy by combining them with nonradioactive drugs inhibiting the key molecular functions within the cancer genome. For example, many low-molecular-weight compounds have been developed as inhibitors that block specific molecules and their pathways in the cancer genome [49]. A combination of *β^−^*/*α*-particle TRT with such inhibitors could make a substantial difference in their DNA-damaging capability and resultant cytotoxicity in cancer cells compared to normal cells.

Hence, there are possibilities for making current TRT more effective and useful. The target choice is important for determining the therapeutic effectiveness of both Auger electron TRT and combination therapy with TRT. The following subsection proposes potential partners and targets within cancer cells, especially related to DNA damage response/repair, as well as to the maintenance systems regulating cell division/death. We also introduce the potential roles of these molecules to be modulated by inhibitors to increase therapeutic efficacy.

### 3.1. DNA Damage Response/Repair Factors Controlling the Cell Cycle

The DNA damage response and repair of cancer cells include a variety of abnormal molecular systems and pathways. These are promising as targets for anti-cancer strategies to selectively induce critical DNA lesions in cancer cells. In this subsection, we will discuss the related enzymes (e.g., PARP, ATM, and ATR) as potential targets and partners in TRT (Figure 2).

#### 3.1.1. PARP

Poly adenosinediphosphate (ADP)-ribose polymerase (PARP) is involved in the genomic integrity of cancer cells and its inhibitors are used as an anticancer strategy [50]. Some inhibitors of PARP (PARPi) have been approved by the FDA as anticancer drugs for breast, ovarian, and prostate cancers that have a DNA repair deficiency (for example, BRCA1/2-deficient cells) [51,52]. Many clinical trials on combination therapies, including a PARPi, are ongoing for a variety of cancer types with different treatment histories [53]. Considering that PARPi potentially blocks DNA damage repair, PARPi would be a promising partner/target for TRT.

PARP-1 and 2 are enzymes that play important roles in DNA damage response and repair. In particular, PARP-1 works in the process of the base excision repair (BER) of DNA single-strand breaks (SSBs). PARP-1 is recruited to SSB sites, and polymers of ADP-ribose (poly(ADP-ribose)) are attached to PARP-1 itself, DNA-repair-associated proteins, and histones while using nicotinamide adenine dinucleotide (NAD^+^) as a substrate [50,53]. This translational modification process is called poly(ADP-ribosyl)ation and promotes repair responses such as BER. PARP is also involved in repairing DNA double-strand breaks (DSBs) via non-homologous end joining (NHEJ) and homologous recombination (HR) [53]. Additionally, poly(ADP-ribosyl)ation itself epigenetically regulates genomic methylation and modulates the chromatin structure, which is related to several cellular processes, including transcription and replication [54,55,56]. Hence, PARP-1 is a key molecule for maintaining several genomic functions, including DNA damage response/repair.

The PARP-1 molecule consists of three main domains: (1) a catalytic domain that catalyzes enzymatic activity, cleaving NAD^+^ into nicotinamide and ADP-ribose; (2) a DNA-binding domain containing three zinc finger motifs; and (3) an auto-modification domain that functions as the target of poly(ADP-ribosyl)ation [50,53]. After PARP-1 binds to damaged sites of DNA via the zinc fingers, NAD^+^ binds to the catalytic domain of PARP-1, leading to conformational changes in the PARP-1 and DNA complex [50,53]. The binding structure and affinity of PARP-1 to DNA are crucial for the subsequent repair processes to proceed. Current PARPi (olaparib, niraparib, rucaparib, and talazoparib, etc.) mimic the nicotinamide moiety in NAD^+^ and block its binding pocket in the catalytic domain. Due to this inhibition of catalytic function, PARP cannot work properly on DNA, resulting in errors in DNA repair and other PARP-associated epigenetic functions.

Based on the pharmacology of the PARPi inhibition of DNA damage repair, it would be reasonable to combine PARPi with DNA-damaging therapeutic modalities. Radiotherapy, including TRT, causes its therapeutic effects mainly by inducing extensive DNA damage [57]. The irradiation of cells is random and not uniform in radiotherapy, which results not only in DSBs, but also SSBs, base damage, and DNA–protein cross-links [58,59,60]. Several studies have demonstrated that combination therapy with PARPi increases the number of DSBs by inhibiting damage repair [61,62,63,64,65,66], thereby enhancing radiation sensitivity. In the setting of TRT, recent studies also have demonstrated that PARPi can enhance the anti-tumor activity of [^177^Lu]Lu-DOTATATE [67,68]. Such combination approaches of DNA damage repair inhibition with TRT are becoming more common these days [68].

In addition, PARPi themselves have been used as molecular-targeting agents for TRT. PARPi can bind to the PARP proteins on DNA without inhibiting the DNA binding of PARP itself. Such indirect DNA-binding of PARPi means that PARPi would be close to DNA in the cell nuclei. In recent years, several research groups have developed different PARPi labeled with a variety of radionuclides, including ^211^At [24,69,70], ^123^I [71,72], and ^125^I [24,25,73,74,75]. In some cases, therapeutic effects have been obtained to a certain degree. Additionally, damage by radiation to PARP-assembled DNA damage sites may create critical lesions that lead to cell death.

In the case of Auger electron emitters, PARPi can be used as a scaffold to deliver the radionuclide to key sites on DNA molecules. However, to date, this approach for inducing critical DNA damage with Auger emitters has not been as effective as hoped. Whether it is sufficient for Auger-electron-emitting radionuclides to bind to DNA indirectly via a PARP protein intermediary located on DNA molecules remains to be demonstrated. First, PARP and its associated proteins might attenuate the physiochemical damage caused by the Auger electrons. Second, particularly for very-low-energy Auger electrons, not being directly bound to DNA increases the distance from the site of decay to the DNA, and such geometrical cushioning may affect the efficiency—this needs to be investigated. Another concern is whether the number of radionuclides that can be delivered to DNA damage sites when targeting PARP with PARPi-labeled agents is enough for the therapy to be effective; this needs to be clarified. Since PARP indeed recognizes DNA damage and could be recruited to those sites, more damage and PARP molecules could then deliver more Auger-electron-emitting PARPi to the cancer genome. Therefore, it could be prudent to evaluate in advance of the treatment if sufficient PARP (hyper)activity occurs to warrant such treatment.

For *α*/*β*^−^-particle TRT, the range of *α*-particles (40–100 μm) and *β^−^*-particles (0.05–12 mm) compared with the dimensions of proteins [2] makes it difficult to transfer energy to DNA molecules efficiently and specifically. Therefore, *α*/*β^−^*-particle-emitting agents might gain little, if any, advantage in targeting and damaging DNA via PARP complexation by radiolabeled PARPi. A possible exception is the high-LET damage that could be created by the recoil nucleus emitted during *α*-decay, which has a tissue range of <0.1 μm. In addition, if a PARP-targeted agent affects the retention in cancer cells via an upregulation of PARP expression, this system also could be a potential targeting strategy for *α*/*β*^−^-particle TRT.

#### 3.1.2. ATM and ATR

Ataxia telangiectasia mutated (ATM) is a kinase that plays a central role in the DNA damage response (DDR) and cell cycle regulation following DNA damage, especially DSBs. Following DSBs, a protein complex consisting of MRE11, RAD50, and NBS1 (MRN complex) binds to the collapsed fork. ATM is activated and recruited to the sites bound to the MRN complexes [76,77]. ATM then phosphorylates several key proteins, including tumor suppressors such as p53, CHK2, and BRCA1. This initiates the activation of the DDR and cell cycle checkpoint, leading to cell cycle arrest, DNA repair, or cell death.

ATM kinase inhibitors (ATMi) may be promising candidates for use as a partner in TRT. Epigenetic defects of the ATM gene have been observed at a high rate in a variety of cancer cells [78,79,80,81]. The upregulated ATM gene is thought to be related to the resistance to DSBs in some radiation/drug-tolerant cells [82,83]. Therefore, for DSB-tolerant cancer cells with upregulated ATM signaling, combination therapy with an ATMi could make TRT more effective than TRT alone by increasing the number of unrepaired DSBs. Although ATMi have yet to be evaluated at the clinical level to the extent of PARPi, the basic mechanism for targeting DDR proteins on DNA damage sites is the same for both ATMi and PARPi [77]. In particular, DSBs are known to be critical damage events that cause cell death, and the inhibition of ATM-initiating DSB repair could generate more lesions in the cancer genome. Several inhibitors (AZD0156, AZD1390, M4076, M3541, and XRD-0394, etc. [84]) are undergoing clinical evaluation. We should keep a close eye on the results of these trials and, in particular, whether they support the concept of combining ATMi with TRT.

Moreover, the safety, effectiveness, and risks of ATMi need to be evaluated carefully and better understood in the future. The ATM gene severely affects DDR; however, its deletions and mutations are known to increase the risk of cancer [85,86]. Thus, the defects and downregulation of the ATM gene in normal cells might cause unwanted adverse effects and risks, and the proper drug delivery system would need to be identified and developed.

Ataxia telangiectasia and rad3-related (ATR) is a serine/threonine-specific protein kinase that also is involved in sensing DNA damage [87]. The ATR-associated pathway is a major DDR route for SSBs. DNA replication stress and DNA damage (SSBs and DSBs, etc.) primarily generate an exposed single-stranded DNA structure (ssDNA). ATR is specifically activated in response to ssDNA coated with replication protein A (RPA) [87]. After its activation, ATR phosphorylates checkpoint kinase 1 (CHK1). CHK1 coordinates the DNA damage cell cycle checkpoint, properly leading to cell death, cycle arrest, or repair [87]. Thus, ATR also plays a crucial role in early DDR and is required as well as PARP and ATM for the repair of the damaged sites on DNA molecules. Compared to ATMi, more ATR inhibitors (ATRi) have been developed and investigated in clinical trials. Examples include Berzosertib [M6620], Ceralasertib [AZD6738], M4344, and BAY1895344 [84]. Some of these clinical studies have demonstrated that ATRi were more effective than PARPi in ATM-deficient tumors [88,89]. For example, a recent clinical study using BAY1895344 demonstrated high antitumor activity in various advanced solid tumors, particularly those with deleterious ATM mutations and/or loss of the ATM protein [90].

It is important to appreciate that, due to tumor heterogeneity, there should be a wide range of profiles in DDR among patients. Ideally, it will be important to select a therapy that is appropriate for the DDR characteristics of each patient’s tumor. Molecular imaging with appropriately labeled DDR inhibitors might be a useful tool for accomplishing this.

For both ATM and ATR, it is unclear whether such proteins and their inhibitors can be used as targets/targeting agents for Auger electron therapy. As described above, the functions of ATM/ATR are crucial in several processes (e.g., DNA damage response/repair, cell cycle checkpoints) that are essential for cell survival. However, ATM and ATR do not sense DNA damage directly, but are instead recruited to the DNA–protein complexes on damaged sites secondarily. Both ATM and ATR are expected to bind to DNA molecules indirectly via multiple protein complexes [91,92]. It is thus unclear whether the resultant distance between the DNA and Auger-electron-emitting radionuclide decay site will adversely affect the therapeutic results. This concern applies to several other downstream DDR proteins associated with ATM/ATR, such as CHK1/CHK2, WEE1, and CDK1/CDK2/CDK4/6, which are known to lead to the activation of cell cycle checkpoints and DNA damage repair. For example, ATR/ATM activates downstream effector kinases such as CHK1/CHK2 in the ATR and ATM-associated DDR pathways, and CHK1/2 inhibitors also are promising anticancer drugs [77,84,87]. Thus, a variety of proteins are required around DNA-damaged sites, but they bind indirectly or sometimes do not bind to DNA molecules at all. Indeed, the practical aspects of Auger electron TRT are largely unknown (the number of decays required for cell killing, relationship between the range, energy deposition, and efficiency of cytotoxicity, and dose rate effects, to name a few). Lacking this information, it is difficult to predict a priori whether these molecules that do not bind directly to DNA are potential targets for Auger electron therapy or not. As is generally the case, experimental evidence is likely to answer these questions earlier than theoretical predictions. In contrast, these inhibitors in combination therapy with TRT as a partner are likely to enhance therapeutic effectiveness by inhibiting the response/repair to TRT-induced DNA damage.

#### 3.1.3. DNA-PK, LIG4, and XRCC4

The DNA-dependent protein kinase (DNA-PK) consists of a catalytic kinase subunit (DNA-PKcs) and the heterodimer DNA end-binding complexes Ku70 and Ku80 [93,94]. DNA-PK plays an upstream role in repairing DNA via nonhomologous end joining (NHEJ) for DSBs [93,94]. DNA-PK phosphorylates the proteins that are involved in DDR/cell cycle checkpoint/DNA damage repair [93,94]. Several inhibitors, including MSC2490484A [M3814], AZD-7648, CC-115, BR2002, and BR101801, are currently undergoing clinical investigation [84]. Many human cancer types have increased DNA-PK expression, which is correlated with a poor patient prognosis [95,96]. Hence, DNA-PK could be worthy of consideration as a potential target/partner for TRT.

X-ray repair cross-complementing 4 (XRCC4) and DNA ligase 4 (LIG4) are downstream proteins in DNA-PK-initiating NHEJ [97]. XRCC4 and LIG4 have important functions in NHEJ. Because they are recruited to the damaged sites, they are also target/partner candidates, albeit with the same concerns noted for the other possibilities introduced above.

#### 3.1.4. Others

The spindle assembly checkpoint (SAC, also known as the mitotic checkpoint) operates as a cell cycle checkpoint during mitosis or meiosis [98,99]. SAC is composed of several protein complexes: MAD1–3, BUB1–3, and MPS1 [98,99]. Monopolar spindle 1 (MPS1, also known as TTK protein kinase) and budding uninhibited by benzimidazoles 1 (BUB1) are key regulators of the SAC [98,99]. MPS1 and BUB1 are overexpressed in several cancers and their inhibitors are clinically emerging in cancer therapy [98,99]. SAC-related inhibitors have been combined with the antimitotic compounds docetaxel and paclitaxel [100]. Although combination therapies with radiation including conventional radiotherapy have yet to be evaluated, clinically effective inhibitors might help to increase the therapeutic effectiveness of TRT. As with other candidates excluding PARPi, as discussed earlier, whether Auger electrons can effectively damage SAC-associated factors on a larger scale of chromosomes has not been investigated.

### 3.2. Amplified and Overexpressed Factors Regulating the Cancer Genome

The unimpeded proliferation of cancer cells is derived from the abnormal transcription and epigenetic modification in the cancer genome that is based on genomic instability [101,102,103]. In most cases, there are mutations in the genes that are important for normal function or in the suppressor genes that inhibit malignant transformation. Other cancer-related genes are upregulated instead, which drives infinite DNA replication and cell division. In particular, enhanced transcription factors and epigenetic regulators have wide-ranging functions that are definitively linked with DDR. For example, PARP-1 is known to be a key molecule not only in DNA repair, but also in chromatin modification and transcriptional regulation [54,55,56]. Although not all key onco-factors recognize/repair DNA damage directly, they help to maintain the cancer genome while escaping critical DNA damage and cell death. Therefore, they could be promising molecular targets in future TRT strategies.

The downregulation of such gene amplification and overexpression with inhibiting agents disables the normal functions in cancer cells, wherein the survival mechanism depends on the gene [104]. In that situation, cancer cells would not proceed normally with replication and division, and their DNA damage response and repair capacity also would be reduced, which could potentially deliver a specific damaging effect to cancer cells. Additionally, while several small-molecule inhibitors alone have a moderate therapeutic efficacy, they are much more effective in combination therapies with immune checkpoint inhibitors [105,106,107]. This suggests that damaging the core onco-factors in the cancer genome should be compatible with immunotherapy. If anticancer immunity is activated efficiently, the therapeutic effect could be expected even in the tumor areas that are not reached by the radiation from the TRT drug. Hence, the upregulated core onco-factors that are related to abnormal transcription and epigenetic modification might be promising targets for Auger electron therapy, and their inhibitors have the potential to be partners for TRT. We shall now discuss the target candidates that are related to the transcription and epigenetic modification of cancer cells.

#### 3.2.1. Transcription-Related Factor

Many transcription-related genes are amplified and overexpressed in cancer cells as a consequence of their malignancy [103,104]. There is a wealth of evidence indicating that transcription-related oncogene expression is correlated with a poor patient prognosis. Notable examples include runt-related transcription factors (RUNXs) in several cancer cell types [108] (RUNX1: acute myeloid leukemia, gastric cancer, hepatocellular carcinoma, and breast cancer [109,110,111,112]; RUNX2: osteosarcoma, bone metastasis of prostate cancer, breast cancer, and so on [113,114,115]; and RUNX3: epithelial ovarian cancer and basal cell carcinoma) [116], N-Myc (MYCN) in neuroblastoma [117], and C-Myc (MYC), which contributes to the cause of at least 40% of human tumors [118]. Additionally, several core oncogenes regulate and activate associated gene expression [103,104]. Considering that the downregulation of such core genes can also decrease the expression of other crucial genes, the unstable molecular networks centered on the key onco-genes can support cancer cell survival and escape cell death [103,104]. In addition to transcription factors, there are many other genes that can be amplified and overexpressed in cancer cells (for example, the androgen receptor gene in prostate cancer cells [119]), and they play an important role in gene expression.

Let us consider the oncogene MYCN, which is a transcription factor that is amplified in human neuroblastoma and is related to its prognosis [117]. The survival hallmark of MCYN-amplified cells is MYCN-related transcription [117,120]. Kang et al. demonstrated that the RNA interference targeting of MYCN-inhibited cell proliferation caused cell death in MYCN-amplified cells more than in non-amplified cells [121]. Additionally, Durbin et al. showed that MYCN, HAND2, ISL1, PHOX2B, GATA3, and TBX2 are members of the transcriptional core regulatory circuitry (CRC) that maintains the cell state in MYCN-amplified neuroblastoma [122]. Furthermore, many researchers have found a relationship between MYCN gene expression and several DDR factors, including PARP [123], ATM [124], and so on [125]. Thus, like MYCN, amplified/overexpressed transcription-related factors—along with some associated components—maintain the cancer genome.

Targeting such core oncogenes with Auger-electron-emitting radiotherapeutics could be a promising strategy for making the damage caused by Auger electrons more effective. A general survey of the literature did not find any publications on gene-targeted Auger electron therapy. However, Watanabe et al. reported that ^111^In-labeled N-myc antisense oligonucleotides targeting the mRNA of MYCN-amplified neuroblastoma cells could delay cell proliferation [126]. Direct damage to genes rather than mRNA is expected to be a more effective therapeutic strategy due to the low concentration (<<1 nmole) of the radioactive compound and the nano-scale range of its Auger electrons. To deliver radionuclides to the target gene, sequence-specific DNA binding molecules such as gRNA (widely used in the CRISPR-Cas9 technique [127]) or pyrrole–imidazole polyamides (PIPs) (used in preclinical cancer therapy studies in mice and marmosets [128,129]) would be available. Additionally, the knockdown of key oncogenes by RNA interference or small inhibiting compounds could be helpful in combination therapy with TRT. In summary, it might be therapeutically advantageous to disrupt the normal state of the cancer genome and cause radiation-induced DNA damage at the same time.

#### 3.2.2. DNA Methyltransferase and Epigenetic Regulators

Cancer cells have accumulative epigenetic alterations, which change the gene activation or silencing functions and enable their dysregulated proliferation [130,131,132]. Specific alternations of DNA methylation and histone modification have been observed at the clinical level in a variety of cancer cell types [133]. They contribute to abnormal gene expression and maintain the cancer genome while escaping cell death during their rapid replication. If the number of genes for an epigenetic modification factor is amplified, it could serve as a potential target for Auger electron therapy. Overexpressed epigenetic modification enzymes that can bind to DNA are expected to play a key role in creating a sustainable genome and also would be target candidates for Auger electron therapy. A variety of inhibitors for cancer epigenetic regulators have been developed and evaluated at the clinical level [134]. For example, many DNA methyltransferase inhibitors (DNMTi) and histone deacetylase inhibitors (HDACi) are approved by the FDA [135,136]. These drugs might be great partners in combination therapy approaches with TRT.

In epigenetic therapy, damaging these factors and regulators not only induces cancer cell death, but also activates the antitumor immune system, which should enhance the response to immunotherapy [105,106,107,137]. This suggests the possibility of developing radioactive analogues that both damage DNA and activate anti-tumor immunity. For example, the DNA methyltransferases SUB39H1 [138] and SETDB1 [139] are amplified/overexpressed in several cancers. They are involved in the silencing of repetitive sequences and escaping cell death. Their amplification/overexpression is correlated with a poor patient prognosis and low response rate to immunotherapy. Interestingly, treatment with an SUB39H1 inhibitor (F5446) and SEDB1 inhibitor results in a superior therapeutic efficacy when combined with immunotherapy. This likely reflects the activation of the cyclic GMP–AMP synthase stimulator of the interferon genes (cGAS–STING) pathway [138,139].

Thus, the downregulation of DNA methyltransferases and epigenetic regulators can potentially activate immunity and increase the response to immunotherapy [138,139,140]. In recent years, the cGAS-STING pathway has been attracting interest in cancer therapies involving DDR and the natural antitumor immune system [141]. Several inhibitors in epigenetic therapy could be capable partners with TRT, causing both therapeutic effects and immune activation. Additionally, Auger-electron-emitting radiotherapeutics that target the core DNA methyltransferases or their genes might provide a therapeutic efficacy derived from DNA damage and also activate immune-related pathways such as the cGAS-STING pathway.

## 4. Perspectives: Possible Limitations

Finally, we shall introduce the potential limitations of these strategies, as well as further aspects that should be considered in future research. The presented targeting strategy for Auger electron therapy and the proposal of combination therapy with TRT are based on the dysregulated DNA damage response/repair ability and altered transcription/epigenetic system in cancer cells. Their functions are wide-ranging due to tumor heterogeneity and related factors create complex molecular networks. Therefore, their therapeutic success or failure would depend on whether the cancer cells being treated are highly reliant on the targeted molecular mechanism in question. Despite having average tendencies, the genomic profile of cancer cells is quite variable, and there is no molecular mechanism that can be applied to all cancers, even for a given disease type. Therefore, in order to increase the likelihood of success, the genetic profile required for the treatment for each therapeutic mechanism should be identified to the greatest extent possible. In the clinical context, in order to have a good response, the screening of patients, perhaps using an in vitro test or molecular imaging, would definitely be essential.

Additionally, there are many radionuclides that emit Auger electrons at different energies and numbers of emissions per decay [142,143]. As the atomic number increases, the numbers of Auger electron emissions and their energies also tend to increase. Therefore, the optimal drug designs suitable for efficiently delivering specific damage to targeted molecules may differ among radionuclides with different effective tissue ranges. Although a rigorous discussion based solely on such Auger electron spectrum data from simulations is difficult, the suitability of radionuclides for specific molecular targeted therapeutic strategies will be an important consideration for Auger electron TRT.

## 5. Conclusions

Focusing on the cancer-specific dysregulated molecular systems that are related to DNA damage response/repair and transcription/epigenetics, we summarized and discussed the potential targets and partnering approaches that might enhance the effectiveness of TRT in the future. Key molecules, including PARP, ATM/ATR, amplified/overexpressed transcription factors, and DNA methyltransferases, maintain the cancer genome and keep abnormal DNA replication and cell division operating, while escaping critical DNA damage and cell death. These crucial molecular factors have the potential to be valuable targets for Auger electron therapy. In addition, non-radioactive inhibitors of these molecular targets could be combination therapy partners to enhance the therapeutic response of TRT. We hope that these tactics will help in overcoming the several limitations in current TRT, including radiation/drug-tolerant cells, tumor heterogeneity, and response rate. By addressing these issues using the strategies proposed herein, it might be possible for TRT to realize its potential and become an essential tool for the treatment of patients with cancer.

## Figures and Tables

**Figure 1 pharmaceutics-15-01926-f001:**
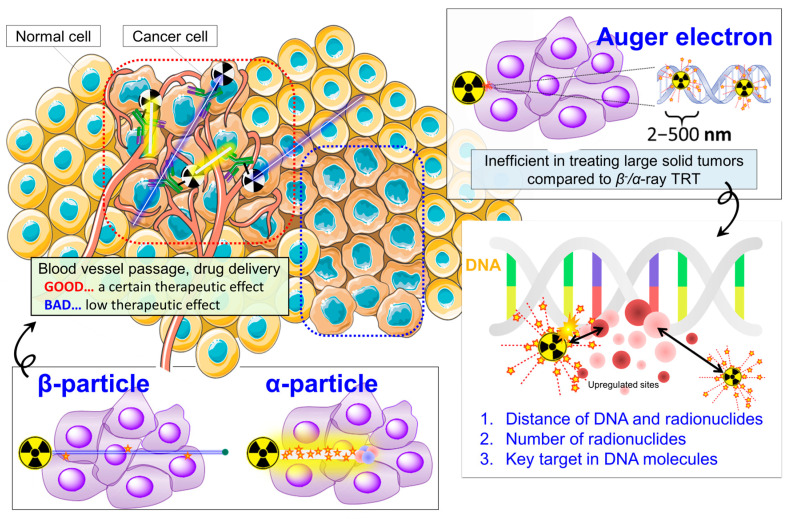
Considerations for TRT with radionuclides emitting *α*-particles, *β*^−^-particles, and Auger electrons.

**Figure 2 pharmaceutics-15-01926-f002:**
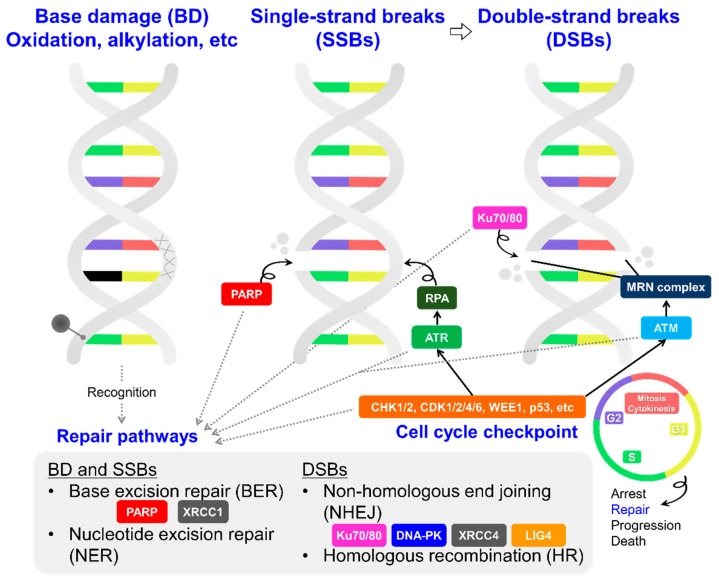
Key molecules in DNA damage response and repair pathways that could be targeted in future TRT strategies.

**Table 1 pharmaceutics-15-01926-t001:** Representative studies of Auger-electron-emitting radiopharmaceuticals for TRT.

Stage of Investigation	Category	Drug	Cancer Type	Year	Country	Notes	Reference
Clinical (adjuvant use)	antimetabolite	5-[^125^I]iodo-2’-deoxyuridine	Recurrent and advanced pancreatic cancer relapsed with cytologically proven neoplastic meningitis	2008	USA	Intrathecal administration as adjuvant treatment (1850 MBq). Feasible without acute neurological toxicity. Seemed to have produced a biological response.	[16]
Clinical (dose escalation study)	antibody	^125^I-labeled mAb A33	Advanced chemotherapy-resistant colon cancer	1996	USA	Phase 1/2 study to determine the maximum tolerated dose (21 patients with doses up to 350 mCi/m^2^). No bowel or bone marrow toxicity, modest antitumor activity.	[17]
Clinical (adjuvant use)		^125^I-labeled MAb 425 (targeting EGFR)	High-grade glioma	1990~2004	USA	Significant increase in median survival as an adjuvant treatment following initial surgery and postoperative external beam radiation therapy	[18]
Clinical (dose escalation study)	somatostatin receptor targeted peptide	[^111^In-DTPA^0^]octreotide	Somatostatin receptor-positive neuroendocrine tumors	2002	The Netherlands	50 patients (cumulative doses of 20 to 160 GBq). Therapeutic effects were seen in 21 patients: partial remission (1), minor remissions (6), and stabilization of previously progressive tumors (14).	[19]
Clinical (dose escalation study)		^111^In-labeled pentetreotide	Progressive, disseminated, and unresectable neuroendocrine tumor (stage III and IV)	2012	USA	112 patients, 500 mCi (18.5 GBq) per cycle and ~4 cycles. Majority (85%) of patients had stable disease. No significant acute toxicity, but grade II/III toxicity was observed.	[20]
Clinical (dose escalation study)	epidermal growth factor	[^111^In]In-DTPA-hEGF	EGFR-positive metastatic breast cancer	2014	Canada	16 patients with doses of up to 2290 MBq. No objective antitumor responses; most common adverse events were flushing, chills, nausea, and vomiting occurring during or immediately post-injection.	[21]
Preclinical (animal)	modular nanotransporter (MNT) targeting EGFR	[^111^In]In-NOTA-MNT	EGFR-overexpressing tumor xenograft (human bladder cancer)	2018	Russia	Significant dose-dependent tumor growth delay (up to 90% growth inhibition) after local infusion of ^111^In-NOTA-MNT in EJ xenograft-bearing mice.	[22]
Preclinical (animal)	antibodies (35A7 mAbs: anti-CEA; m225 mAbs: anti-HER2)	^125^I-labeled 35A7 mAb (non-internalizing), ^125^I-labeled m225 mAb (internalizing)	Vulvar squamous carcinoma cell line A-431 tumor xenograft	2009	France	Compared with unlabeled mAb, non-internalizing ^125^I-labeled 35A7mAb had a significant increase in survival.	[23]
Preclinical (animal)	PARP inhibitor	[^125^I]PARPi-01	Triple-negative breast cancer cell line MDA-MB-231 tumor xenograft	2022	Germany, The Netherlands	Endogenous therapy induced a significant delay in tumor growth. No significant survival advantage, but significantly higher apoptosis ratio, no radiotoxicity in liver or thyroid.	[24]
Preclinical (cell)	PARP inhibitor	^125^I-labeled KX1	High-risk neuroblastoma cell line	2020	USA	^125^I-labeled KX1 was approximately twice as effective as ^131^I-labeled KX1	[25]

## Data Availability

Not applicable.

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
