# Peer review of "DNA Repair Inhibitors: Potential Targets and Partners for Targeted Radionuclide Therapy"

_pharmaceutics, 2023, doi:10.3390/pharmaceutics15071926_

Round 1

Reviewer 1 Report

This is a very interesting and well-written review that contains an expert and detailed analysis of the status of targeted radionuclide therapy (TRT) and of the problems that still afflict this therapeutic approach, and which have so far prevented definitive successes in the fight against cancer. The main merit of this work is to make a detailed scientific proposal aimed at combining TRT with inhibitors of biomolecular processes and strategies that cancer cells use to remedy the damage induced to their genetic material by Auger and alpha radiation or other therapeutic agents. Authors offer a snapshot of various biomolecular pathways that could be, at least in principle, utilized to amplify the effectiveness of TRT. The authors' fundamental approach is undoubtedly fascinating and in line with the current paradigm of targeted molecular therapy. The scientific quality of this article is quite high and justifies its publication. Despite this, an unpleasant feeling that is difficult to escape from reading these types of articles is that it is not clear how the extreme complexity of cellular processes expressed at the molecular level and the multitude of closely linked biomolecular events can be really truly understood and thoroughly checked and, therefore, used for definitive cancer therapy. The history of science teaches that, sometime, remedies taken from nature can prove to be more effective and simpler than those developed after a very fatiguing and often elusive, fundamental biomolecular research (notably, the examples of the iodine-131 anion, and recently of the copper-64 cation).

Author Response

Reviewer Number 1

This is a very interesting and well-written review that contains an expert and detailed analysis of the status of targeted radionuclide therapy (TRT) and of the problems that still afflict this therapeutic approach, and which have so far prevented definitive successes in the fight against cancer. The main merit of this work is to make a detailed scientific proposal aimed at combining TRT with inhibitors of biomolecular processes and strategies that cancer cells use to remedy the damage induced to their genetic material by Auger and alpha radiation or other therapeutic agents. Authors offer a snapshot of various biomolecular pathways that could be, at least in principle, utilized to amplify the effectiveness of TRT. The authors' fundamental approach is undoubtedly fascinating and in line with the current paradigm of targeted molecular therapy. The scientific quality of this article is quite high and justifies its publication.

Our response:  Thanks for the positive comments!

Despite this, an unpleasant feeling that is difficult to escape from reading these types of articles is that it is not clear how the extreme complexity of cellular processes expressed at the molecular level and the multitude of closely linked biomolecular events can be really truly understood and thoroughly checked and, therefore, used for definitive cancer therapy. The history of science teaches that, sometime, remedies taken from nature can prove to be more effective and simpler than those developed after a very fatiguing and often elusive, fundamental biomolecular research (notably, the examples of the iodine-131 anion, and recently of the copper-64 cation).

Our response:  We agree that simple ions like the iodine-131 anion and the copper-64 cation sometimes exhibit better results than their labeled compounds targeting specific biomarkers. This is likely because biomolecular systems frequently are too diverse and complex, which sometimes makes molecular targeted therapy uncertain in its current status. If a “naturally targeted” drug works, like the aforementioned radio-ions, that's fine, of course. However, many types of cancer cannot be targeted in this way.

We speculate that in the future, personalized therapy would be based on the molecular characteristics of a cancer, which potentially can be exploited for a  non-invasive and highly effective treatment. Admittedly, this will be require an understanding of a multitude of biological events, with the level of understanding hopefully increasing with time.  We hope this review summarizes some strategies for attempting this, and provides a few insights and ideas for possible paths forward.

Reviewer 2 Report

The topic is very hot in the research network, especially whom are interested in Auger electron therapy.  The authorks gave an excellent overview of state of art of TRT, and professionally underlined the importance of the biomolecular aspects to increase the therapeutic effect of the cell-irradiation. The authors dedicated step by step the key questions and explained the possible answer in nice logical order. Nice overview, must publish it!

Author Response

Reviewer Number 2

The topic is very hot in the research network, especially whom are interested in Auger electron therapy.  The authors gave an excellent overview of state of art of TRT, and professionally underlined the importance of the biomolecular aspects to increase the therapeutic effect of the cell-irradiation. The authors dedicated step by step the key questions and explained the possible answer in nice logical order. Nice overview, must publish it!

Our response:  Thanks for the positive comments!

Reviewer 3 Report

Interesting manuscript. However, the title is somehow misleading, as it deals mostly with Auger electron emitters, and, to a lesser extent, to alpha or beta particles. It should therefore be modified to fit better the content of the manuscript.

Moreover, if DNA repair inhibitors were to be used in combination with TRT, they can also prove useful for beta emitters, as was for instance demonstrated preclinically with different 177Lu-labeled compounds combined with PARP inhibitors.

Nomenclature has to be followed (for instance [177Lu]Lu-DOTATATE). It is sometimes the case, sometimes not. Please homogenize.

Reference part has to be modified. It is sometimes difficult to read, when multiple references are indexed under the same number (sometimes a lot, i.e. Ref. 18).

Otherwise the manuscript is well written and well documented and argued. It might prove useful for the advancement of TRT.

Author Response

Reviewer Number 3

Interesting manuscript. However, the title is somehow misleading, as it deals mostly with Auger electron emitters, and, to a lesser extent, to alpha or beta particles. It should therefore be modified to fit better the content of the manuscript.

Our response:  We agree and have modified the title accordingly.

Moreover, if DNA repair inhibitors were to be used in combination with TRT, they can also prove useful for beta emitters, as was for instance demonstrated preclinically with different 177Lu-labeled compounds combined with PARP inhibitors.

Our response:  We agree and have modified the title to reflect this.  In addition, we have added a few lines and references about the combination approaches of TRT with PARPi (page 15, 34-36).

Nomenclature has to be followed (for instance [177Lu]Lu-DOTATATE). It is sometimes the case, sometimes not. Please homogenize.

Our response:  Nomenclature for the radioactive compounds have been checked and homogenized.

Reference part has to be modified. It is sometimes difficult to read, when multiple references are indexed under the same number (sometimes a lot, i.e. Ref. 18).

Our response:  Sorry for creating a chaotic reference list.  It has been modified and hopefully, is easier to read.

Otherwise the manuscript is well written and well documented and argued. It might prove useful for the advancement of TRT.

Our response:  Thanks for the positive comment.